Diet and kwashiorkor: a prospective study from rural DR Congo

Kismul Hallgeir hallgeir.kismul@cih.uib.no
Van den Broeck Jan
Lunde Torleif Markussen
Centre for International Health, University of Bergen , Norway
Pérez-Jiménez Jara
Electronic publication date: 2014 Apr 15
Publication date: 2014
Volume: 2
Electronic Location ID: e350
Received 2013 Nov 22; Accepted 2014 Mar 27
Copyright: © 2014 Kismul et al.
Copyright year: 2014
Copyright holder: Kismul et al.
License: This is an open access article distributed under the terms of the Creative Commons Attribution License, which permits unrestricted use, distribution, and reproduction in any medium, provided the original author and source are credited.
License URL: https://creativecommons.org/licenses/by/3.0/

Keywords: Malnutrition, Kwashiorkor, Marasmus, Food items, β-carotene, Papaya, Sweet potato, Wild vegetables, DR Congo

Funding: University of Bergen and the Nutricia Research Foundation The study was funded by the University of Bergen and the Nutricia Research Foundation. The funders had no role in study design, data collection and analysis, decision to publish, or preparation of the manuscript.

==============================
The etiology of kwashiorkor remains enigmatic and longitudinal studies examining potential causes of kwashiorkor are scarce. Using historical, longitudinal study data from the rural area of Bwamanda, Democratic Republic of Congo, we investigated the potential causal association between diet and the development of kwashiorkor in 5 657 preschool children followed 3-monthly during 15 months. We compared dietary risk factors for kwashiorkor with those of marasmus. Kwashiorkor was diagnosed as pitting oedema of the ankles; marasmus as abnormal visibility of skeletal structures and palpable wasting of the gluteus muscle. A 24-h recall was administered 3-monthly to record the consumption of the 41 locally most frequent food items. We specified Hanley–Miettinen smooth-in-time risk models containing potential causal factors, including food items, special meals prepared for the child, breastfeeding, disease status, nutritional status, birth rank, age, season and number of meals. Bayesian Information Criteria identified the most plausible causal model of why some children developed kwashiorkor. In a descriptive analysis of the diet at the last dietary assessment prior to development of kwashiorkor, the diet of children who developed kwashiorkor was characterized by low consumption of sweet potatoes, papaya and “other vegetables” [0.0% , 2.3% (95% CI [0.4, 12.1]) and 2.3% (95% CI [0.4, 12.1])] in comparison with children who did not develop kwashiorkor [6.8% (95% CI [6.4, 7.2]), 15.5% (95% CI [15, 16.1]) and 15.1% (95% CI [14.6, 15.7])] or children who developed marasmus [4.5% (95% CI [2.6, 7.5]) 11.8% (95% CI [8.5, 16.0]) and 17.6% (95% CI [13.7, 22.5])]. Sweet potatoes and papayas have high β-carotene content and so may some of “the other vegetables”. We found that a risk model containing an age function, length/height-for age Z-score, consumption of sweet potatoes, papaya or other vegetables, duration of this consumption and its interaction term, was the most plausible model. Among children aged 10–42 months, the risk of developing kwashiorkor increased with longer non-consumption of these foods. The analysis was repeated with only children who developed marasmus as the reference series, yielding similar results. Our study supports that β-carotene may play an important role in the protection against kwashiorkor development.

Introduction

Malnutrition contributes significantly to the high under-five year mortality in the world and as an underlying factor it has been estimated that it contributes to over one third of all child deaths (Lim, 2012). Mortality is very high among children with marasmus, and even higher among those with kwashiorkor (Briend, Wojtyniak & Rowland, 1987; Prudhon et al., 1997). Kwashiorkor and marasmus are characterised by different metabolic response to severe undernutrition (Badaloo et al., 2006; Jahoor et al., 2008).

Kwashiorkor has been linked to diet since its first description. Williams (1935), who introduced the name kwashiorkor, suggested protein undernutrition as the etiology of kwashiorkor. The association of kwashiorkor with low protein intake has later been questioned. So far no research has demonstrated that children with kwashiorkor consume less protein than children with marasmus. Golden & Ramdath (1987) proposed excess free radicals as the explanation of clinical findings in kwashiorkor. Relations between antioxidant depletion and the occurrence of kwashiorkor have been investigated, but the role of oxidative stress as primary cause of kwashiorkor is still debated (Ciliberto et al., 2005; Lenhartz et al., 1998; Manary, Leeuwenburgh & Heinecke, 2000). We would like to remark that there are difficulties with the oxidative hypothesis. As an example oxidative stress is present in HIV but studies have found oedematous malnutrition to occur in a minority among HIV seropositive patients who are severely malnourished (Asafo-Agyei, Antwi & Nguah, 2013).

To our knowledge there are only two observational longitudinal studies that have examined the relations between diet diversity and kwashiorkor. Investigating protein-calorie malnutrition, a study following Indian children from birth to 3 years examined differences in diet between children developing kwashiorkor and children developing marasmus (Gopalan, 1992). A more recent study examined dietary factors determining kwashiorkor by assessing diet of one to three years old Malawian children (Lin et al., 2007). The former study did not find significant differences in diet between children who developed kwashiorkor and those who developed marasmus, while the latter did not find differences between those who developed kwashiorkor and those who did not (Gopalan, 1992; Lin et al., 2007).

The overall aim of our study is to investigate, in a large longitudinal population-based study, the possible association between diet and the development of kwashiorkor. We also wanted to compare the causal influence of dietary risk factors for kwashiorkor with those of marasmus. We performed this analysis by applying the causal investigation method proposed by Miettinen and the statistical approach by Hanley and Miettinen, using a random sample of person moments from the entire dynamic population as the reference series (Hanley & Miettinen, 2009; Miettinen, 2010).

Method

The Bwamanda study

We did a secondary analysis of the historical data from the Bwamanda study, conducted from 1989 to 1991 in a rural area of the northwest part of the Democratic Republic of Congo, (DRC), located at 19.2 degrees east and 3.2 degrees north. The people of Bwamanda are, up till today, predominantly subsistence farmers and the basic diet consists of mainly of maize, cassava supplemented with fish, vegetables and fruits. Health care in the area is provided by a central hospital and 10 minor health centres with a few of these providing some limited nutritional rehabilitation services. With virtually unchanged living conditions in the study area, the secondary analysis was viewed to be contemporary and relevant.

Study design

The Bwamanda study was a dynamic population study with follow-up including thrice-monthly survey rounds, making up 15 months of follow-up and 6 contacts. At the first round 4 235 preschool children were enrolled and at the last round a total of 5 657 were enrolled. A full description of the study population can be found in Van den Broeck, Eeckels & Vuylsteke (1993). Trained interviewers conducted interviews according to an interviewer’s manual. They determined the children’s age on the basis of birth date noted on children’s road to health chart or on parents’ identity paper or on the basis of an interview using a local events calendar.

Children were examined for kwashiorkor by using the presence of pitting oedema of the feet or ankles as a criterion. All children were examined for marasmus through inspection of abnormal visibility of skeletal structures and by absence or near-absence of palpable gluteus muscle. A locally constructed measuring board was used for measuring the length of children below 24 months, while a microtoise was used for measuring children older than 24 months. In both cases length was measured to the nearest 0.1 cm. A spring scale (CMS weighting equipment) was used to weigh the children to the nearest 100 g. We applied the WHO Child Growth Standard for anthropometric scoring (World Health Organization, 2006). Z-scores were calculated for weight for length/height (WHZ) and for length/height for age (HAZ).

At each contact interviewers undertook face-to-face interviews with the most proximal caregiver of the child, usually the biological mother. The questionnaire included a single non-quantitative 24-h recall with the 41 locally most consumed food items listed and interviewees providing “yes or no” answers to the questions if children had consumed the listed food items during the previous day. The food items had been identified through a pilot study. The interviewees were also asked about number of meals prepared for the families, special meals prepared for the child and breastfeeding.

Statistical method

In an initial descriptive analysis, we tabulated the percentage (95% confidence interval) of individuals eating the different items, grouped by those who developed kwashiorkor, those who did not and those who developed marasmus. We used a two-sample test for equality of proportions to test if the fractions were different.

Here, we were interested in estimation of risks of developing kwashiorkor specific to age, diet, frequency of food consumption, and infectious diseases. We were also interested in the duration of a particular diet; did a child eat a food item at each visit occurring every three months over the last 15 months, or only at, for example, one of the interview rounds prior to developing kwashiorkor. Smooth-in-time hazard functions as proposed by Hanley and Miettinen allow this type of analysis (Hanley & Miettinen, 2009). We specified Hanley–Miettinen smooth-in-time risk models containing all potential causal factors, including food items, special meals prepared for the child, breastfeeding, disease status, nutritional status, birth rank, age, season and number of meals. To select a representative sample from the study population we used the method proposed by Miettinen, and used the whole study population as reference series (Miettinen, 2010). In the analysis we include all new cases of kwashiorkor, but use a representative sample of the non-cases. With a relatively small number of cases, there is little to be gained by letting the number of non-cases become arbitrarily large, having in mind the computational cost of running the model. Results are reported as log-odds ratios (LOR) and risk reductions. In line with this method we took the dataset to consist of 35 person moments (c) where kwashiorkor was observed as the case series, and a representative sample (b) of the infinite number of person moments that constitute the 46 397 person-months in the study base. We use a (b)/(c)-ratio of 150 assuring variances and covariance have minimal errors (less than 1 percentage) compared to using the entire series.

Age function and age as a risk factor

Given that risk is not changing linearly with age, as seen in Fig. 1, we developed an age variable that accounted for nonlinear change in risk. Such transformations are required when risk does not change linearly with age. Accordingly age was included as an independent variable in the model. Based on visual inspection of how kwashiorkor was distributed according to age, a transformation of the age variable was done: (1) fage=aexp−b Age∗exp−aexp−b Age.

To find the parameters a and b, we optimized Eq. (2) with binomial errors using logistic regression. (2) y=a+b∗fage.

The AIC (Akaike Information criteria Information criteria) was used to compare models. To find the parameters which minimized AIC we used an algorithm combining the golden section search and successive parabolic interpolation, an efficient and automated method to find the best model. Here we used the optimize function in R to find the parameters which minimized AIC, resulting in a = 11.55, and b = 0.90 (Brent, 1973).

Figure 1 Three months prevalence of kwashiorkor according to age in months in intervals of six months.

Prevalence of kwashiorkor (y-axis) against age in months (x-axis). Prevalence aggregated by age group. Short vertical blue lines indicate age groups. Black dots indicate age of those children who developed kwashiorkor. Age at first time a child was observed with kwashiorkor was used.

Other risk factors

The risk factors associated with kwashiorkor were defined in two steps; first we specified a log-linear hazard model with binomial errors where the independent variables were all food types, presence or absence of diarrhoea, and stunting and wasting at last visit. We defined time as the natural logarithm of number of months a person had, or had not, consumed a specific food item. We assumed that the food items reported at a given point in time were consumed up until the next contact, with the interview during the current visit providing data on any alterations in the consumption patterns since the previous contact.

Each variable was multiplied with the natural logarithm of time the item had been consumed or not. Next, we used the BIC (Bayesian Information Criteria) implemented in R’s MASS package (stepAIC) to find the most plausible model based on our data; the posteriori most probable candidate model. The fitted candidate model corresponding to the lowest value of BIC is the candidate model corresponding to the highest Bayesian posterior probability.

Based on the selected model we address the risk of developing kwashiorkor given a prior personal profile. We report risk reduction estimates on the basis of profiles.

To test if the model could also explain the difference between subjects who developed marasmus from those who developed kwashiorkor we applied the final selected model, with the same 35 person moments (c2) where kwashiorkor was observed as the case series, but this time with the references constituted by a sample of the infinite number of person moments including 1 173 person-months observed in 372 new cases of marasmus.

Results

Table 1 reports the distribution of age, the HAZ score and the WHZ score for children with different nutritional status. It shows that children with kwashiorkor were younger than children with no kwashiorkor and marasmus. HAZ and WHZ scores were lower in children with kwashiorkor than in children with no kwashiorkor, but HAZ and WHZ scores were lower in children with marasmus than in children with kwashiorkor. Table 2 shows that the proportion of children with diarrhoea and anaemia was significantly higher in children with kwashiorkor and marasmus than in normal children. The percentage of children that were dehydrated was also highest in children with kwashiorkor and marasmus. In addition the table shows that there were significantly more boys than girls with marasmus.

Table 1 Age distribution, length/height-for-age Z-scores (HAZ), and weight-for-length/height Z-score (WHZ) for children with different clinical nutritional status.

Z-scores based on the WHO-2006 Child Growth Standards [17].

	Age in month	HAZ	WHZ	
	Q10	Q50	Q90	Q10	Q50	Q90	Q10	Q50	Q90	
Kwashiorkor	15.9	26.5	38.4	−4.3	−2.3	−0.8	−2.5	−1.0	0.1	
Reference population	7.4	35.9	66.5	−2.9	−1.4	−0.1	−1.4	−0.1	1.2	
Marasmus	10.8	28.8	64.0	−4.5	−2.7	−1.2	−2.8	−1.3	0.1	

Table 2 Disease status (% of children), sex and age distribution at survey round prior to first observation of kwashiorkor or marasmus.

The numbers (n) refer to number of observations.

	Normal	Kwashiorkor	Marasmus	
	n = 20 114	n = 41	n = 451	
Coughing (%)	35.5 (34.8, 36.1)	34.1 (20.5, 50.7)	43.0 (39.7, 48.5)	
Diarrhoea (%)	5.1 (4.8, 5.4)	14.6 (6.1, 29.9)	12.1 (9.4, 15.4)	
Anaemia (%)	17.4 (16.9, 18.0)	39.0 (24.6, 55.5)	23.9 (20.3, 27.9)	
Fever (%)	10.8 (10.3, 11.2)	15.4 (6.4, 31.2)	16.9 (13.8, 20.5)	
Dehydrated (%)	0.4 (0.3, 0.5)	7.3 (1.9, 21.0)	4.9 (3.2, 7.3)	
Sex (% male)	51.2 (49.9, 52.5)	47.7 (32.7, 63.1)	60.7 (55.9, 65.3)	
Age in months. First round
(Q10, Q50, Q90)	6.9, 32.5, 61.1	8.8, 18.1, 29.4	5.3, 24.8, 60.1	

Table 3 reports the consumption of different food items by the children in the survey round prior to the incidence of kwashiorkor. A high proportion of the children had consumed cassava roots, maize and cassava leaves. The proportion who had consumed cassava roots and maize was non-significantly higher for those who developed kwashiorkor, but for cassava leaves the consumption was lowest for the children with kwashiorkor. The proportion of children who had consumed yam, pineapple, citrus, snails, and eggs was non-significantly higher for the children who developed kwashiorkor than for the others. On the other hand the proportion of children with kwashiorkor who had consumed okra, ground nuts, banana, squash, meat, chili, fish and other vegetables was non-significantly lower than for the rest. There were no significant differences in proportion of children who had consumed palm oil between children who developed kwashiorkor and the other children. The diet of children who developed kwashiorkor was characterized by low consumption of sweet potatoes, papaya and “other vegetables” [0.0%, 2.3% (95% CI [0.4, 12.1]) and 2.3% (95% CI [0.4, 12.1])]. In comparison the children who did not develop kwashiorkor had higher consumption of sweet potatoes, papaya and “other vegetables” [6.8% (95% CI [6.4, 7.2]), 15.5% (95% CI [15, 16.1]) and 15.1% (95% CI [14.6, 15.7])]. The children who developed marasmus also had higher consumption of these food items than the children who developed kwashiorkor [4.5% (95% CI [2.6, 7.5]) 11.8% (95% CI [8.5, 16.0]) and 17.6% (95% CI [13.7, 22.5])].

Table 3 Consumption of different food items in the survey round preceding the development of kwashiorkor (n = 37) and non-development of kwashiorkor (n = 8 108) and development of marasmus (n = 374) in children between 6 and 50 months of age.

Food items	Children with kwashiorkor	Children without kwashiorkor	Children with marasmus	Food items	Children with kwashiorkor	Children without kwashiorkor	Children with marasmus	
	% (95% CI)	% (95% CI)	% (95% CI)		% (95% CI)	% (95% CI)	% (95% CI)	
African pear	0.0 (0.0, 8.2)	0.0 (0.0, 0.1)	0.0 (0.0, 1.3)	Okra	2.3 (0.4, 12.1)	4.5 (4.2, 4.8)	5.2 (3.2, 8.4)	
Amaranth	7.0 (2.4, 18.6)	1.8 (1.7, 2.1)	3.1 (1.6, 5.8)	Palm oil	86.0 (72.0, 93.4)	88.8 (88.3, 89.3)	80.3 (75.4, 84.5)	
Aubergine	0.0 (0.0, 8.2)	0.8 (0.7, 0.9)	0.0 (0.0, 1.3)	Papaya	2.3 (0.4, 12.1)	15.5 (15.0, 16.1)*	11.8 (8.5, 16.0)*	
Avocado	0.0 (0.0, 8.2)	0.2 (0.1, 0.2)	0.0 (0.0, 1.3)	Pineapple	4.7 (1.3, 15.5)	1.4 (1.2, 1.6)	1.7 (0.7, 4.0)	
Banana	9.3 (3.7, 21.6)	19.1 (18.5, 19.7)	16.3 (12.5, 21.0)	Powder milk	0.0 (0.0, 8.2)	0.0 (0.0, 0.1)	0.0 (0.0, 1.3)	
Beans	31.1 (30.4, 31.8)	0.4 (0.3, 0.5)	0.0 (0.0, 1.3)	Rice	2.3 (0.4, 12.1)	0.6 (0.5, 0.8)	0.0 (0.0, 1.3)	
Breadfruit	0.0 (0.0, 8.2)	1.1 (0.9, 1.2)	1.7 (0.7, 4.0)	Sesame	0.0 (0.0, 8.2)	0.1 (0.1, 0.2)	0.3 (0.1, 1.9)	
Cassava leaves	76.7 (62.3, 86.8)	79.2 (78.6, 79.9)	70.7 (65.2, 75.6)	Shrimp	0.0 (0.0, 8.2)	0.1 (0.1, 0.2)	0.0 (0.0, 1.3)	
Caterpillars	2.0 (1.8, 2.2)	2.0 (1.8, 2.2)	1.0 (0.4, 3.0)	Snails	2.3 (0.4, 12.1)	1.4 (1.3, 1.6)	1.4 (0.5, 3.5)	
Cassava roots	76.7 (62.3, 86.8)	72.6 (71.9, 73.3)	72.3 (66.9, 77.2)	Soya	4.7 (1.3, 15.5)	5.2 (4.9, 5.5)	5.9 (3.7, 9.2)	
Chili pepper	4.7 (1.3, 15.5)	8.9 (8.4, 9.3)	4.5 (2.6, 7.5)	Spinach	2.3 (0.4, 12.1)	2.8 (2.6, 3.1)	2.1 (1.0, 4.5)	
Egg	4.7 (1.3, 15.5)	0.7 (0.5, 0.8)**	1.0 (0.4, 3.0)	Squash	0.0 (0.0, 8.2)	4.9 (4.6, 5.2)	5.2 (3.2, 8.4)	
Fish	18.6 (9.7, 32.6)	31.1 (30.4, 31.8)	25.3 (20.6, 30.6)	Sugar cane	0.0 (0.0, 8.2)	0.7 (0.6, 0.9)	0.3 (0.1, 1.9)	
Fruit (others)	0.0 (0.0, 8.2)	1.8 (1.6, 2.0)	1.0 (0.4, 3.0)	Sweet potato	0.0 (0.0, 8.2)	6.8 (6.4, 7.2)	4.5 (2.6, 7.5)	
Ground nuts	18.6 (9.7, 32.6)	28.6 (27.9, 29.3)	23.9 (19.3, 29.1)	Termites	0.0 (0.0, 8.2)	0.3 (0.3, 0.4)	0.0 (0.0, 1.3)	
Maize	97.7 (87.9, 99.6)	93.5 (93.1, 93.8)	91.7 (88.0, 94.4)	Tomatoes	0.0 (0.0, 8.2)	1.3 (1.1, 1.5)	0.0 (0.0, 1.3)	
Mango	0.0 (0.0, 8.2)	0.9 (0.7, 1.0)	0.7 (0.2, 2.5)	Wheat	0.0 (0.0, 8.2)	0.6 (0.5, 0.7)	0.3 (0.1, 1.9)	
Meat	0 .0 (0.0, 8.2)	4.7 (4.4, 5.0)	5.5 (3.4, 8.8)	Other vegetables	2.3 (0.4, 12.1)	15.1 (14.6, 15.7)*	17.6 (13.7, 22.5)**	
Milk	0.0 (0.0, 8.2)	0.1 (0.1, 0.1)	0.0 (0.0, 1.3)	Yam	2.3 (0.4, 12.1)	1.3 (1.1, 1.4)	0.7 (0.2, 2.5)	
Mushroom	0.0 (0.0, 8.2)	2.3 (2.1, 2.5)	1.7 (0.7, 4.0)					
Notes.

* denotes p-value <0.05

** denotes p-value <0.01 with the value estimated using 2-sample test for equality of proportions with continuity correction as implemented in the prop test in R.

Given that the β-carotene could be the main acting agent in sweet potatoes, papaya and “other vegetables” we constructed a variable, PaSV (papaya, sweet potato and “other vegetables”), which combined all these items, weighted by the β-carotene content of 100 g of each item. The weighting of sweet potatoes equalled 1 and papaya 1/3. The variable “other vegetables” includes taro, taro leaves and wild vegetables. Taro leaves are rich in β-carotene and a study from DRC shows that wild vegetables are also rich in β-carotene (Termote et al., 2012). The PaSV variable did not encompass cassava leaves and amaranth. Since we have not been able to determine the more precise content of the other vegetables in our study, we have weighted the “other vegetables” low, with the weighing equal to 1/10. For the construction of smooth-in-time risk models we defined time for this combined variable as for the single food items.

The most probable model based on BIC included age, time, PaSV, and HAZ. The two variables were correlated (R2 = 0.50). As seen in Fig. 1 the risk of developing kwashiorkor was highest in the age interval between 16 and 38 months.

Table 4 shows the coefficients for the non-proportional hazard model with person moments sampled from the entire population. The log-odds for the continuous variable HAZ; LOR −0.8 (CI 95% [−1.1, −0.5]), length/height for age Z-score, describes an increased risk of developing kwashiorkor with lower height for age. Chronic malnourished children on average have a negative HAZ score, hence the negative log-odds. We found the log-odds for the time variable to be LOR 4.7 (CI 95% [3.4, 6.1]), for the combined variable for food items containing β-carotene it was LOR −9.2 (CI 95% [−21.0, −3.1]), for PaSV, and their interaction it was LOR 8.1 (CI 95% [−11.1, −2.1]). These findings must be understood together. A child not consuming the PaSV food items will have PaSV = 0, and thus the interaction term is also zero. The risk of developing kwashiorkor therefore increases the longer the child does not consume the PaSV food items. On the other hand, as illustrated in Fig. 2 a child consuming PaSV food items, PaSV >0, will reduce the risk over time. The overall model fit was good with an AIC of 251.3 and a Nagelkerke R2 index of 0.44.

Table 4 The coefficients for the non-proportional hazard model with person moments sampled from the entire population.

The age variable is a transformation based on the distribution of kwashiorkor across age. T is a variable that describes the duration of consuming a food item containing β-carotene. PaSV is a variable that combines papaya, “other vegetables” and sweet potatoes and weighted by the β-carotene content of 100 g of the item. The height-for-age Z-scores (HAZ) are based on the WHO-2006 Child Growth Standards [17].

Term	Log odds—estimate	Confidence interval, 95%	
Intercept	−15.5	−18.1, −13.4	
Age function of age (months)	10.1	6.1, 14.8	
T months	4.7	3.4 , 6.1	
PaSV	−9.2	−21.0, −3.1	
HAZ	−0.8	−1.1, −0.5	
T ∗ PaSV	−8.1	−11.1, −2.1	

Figure 2 Risk reduction for developing kwashiorkor showing reduction of consuming β-carotene rich products according to age in months.

The dotted line is risk reduction after two months, dashed line after four months, and solid line after six months. (A) shows risk reduction for a child with a height-for-age Z-score (HAZ) of minus five, (B) for a child with HAZ of minus three, and (C) a child with HAZ of zero. HAZ-scores are based on the WHO-2006 Child Growth Standards [17].

Table 5 shows the findings from sampling control-moments only from children who developed marasmus. The table shows that the HAZ score for those who developed marasmus is the same as the HAZ score for the children who developed kwashiorkor (LOR = 0.0, CI 95% [−0.1, 0.2]). Then again it shows that there is a difference with regards to consumption of products containing β-carotene with the LOR for PaSV being −6.8 (CI 95% [−17.8, −1.7]) and for PaSV combined with the time variable T the LOR was −6.3 (CI 95% [−9.0, −0.8]). The age of children who developed kwashiorkor was also different from children who developed marasmus with LOR being 7.9 (CI 95% [4.3, 12.1]). The age of children with marasmus was distributed within the age of 11–64 months, while the age of children with kwashiorkor mainly fell between 16 and 38 months, reaching a top around 26 months.

Table 5 The coefficients for the non-proportional hazard model with reference person moments drawn from people developing marasmus.

The age variable is based on the distribution of kwashiorkor across age. T is a variable that describes the duration of consuming a food item containing β-carotene. PaSV is a variable that includes papaya, “other vegetables” and sweet potatoes weighted by the β-carotene content of 100 g of the item. Height-for-age Z-scores (HAZ) are based on the WHO-2006 Child Growth Standards [17].

Term	Log odds—estimate	Confidence interval, 95%	
Intercept	−9.7	−12.1, −7.7	
Age function of age (months)	7.9	4.3, 12.1	
T months	4.2	3.2, 5.4	
PaSV	−6.8	−17.8, −1.7	
HAZ	0.0	−0.3, 0.2	
T ∗ PaSV	−6.3	−9.0, −0.8	

Ethical approval for the Bwamanda study was granted by the University of Leuven’s Tropical Childcare Health Working Group. Community consent was obtained verbally from community leaders, whereas individual verbal consent was obtained from children’s caretakers.

Discussion

Our study shows that the children who developed kwashiorkor were mainly stunted children aged 16–38 months. Their diet was characterized by a low or no consumption of sweet potatoes, papaya and “other vegetables”. The children who developed marasmus resembled the children who developed kwashiorkor by being stunted. However, the consumption of papaya, sweet potatoes and “other vegetables” were lower among children with kwashiorkor than among marasmic children. Sweet potatoes, papaya and “other vegetables” are characterised by their high β-carotene content. We found that the risk of developing kwashiorkor increased the longer the child did not consume these food items.

We emphasize the importance of β-carotene because it is a substance with significant antioxidant activities. Also, sweet potatoes, papaya and what we have termed “other vegetables” contain several other carotenoids that have antioxidant activities. As a result the various carotenoids, the mixture of carotenoids or carotenoids in association with other antioxidants in these food items can have played important roles in the protective pathway in kwashiorkor.

Children who developed kwashiorkor also consumed palm oil, with the oil being rich in β-carotene. In Bwamanda palm oil is extracted locally and mostly stored in bottles, often being exposed to strong sunlight. Palm oil is used for preparing cassava leaves stew. The oil is mixed with cassava leaves and boiled in a pot for an hour. Studies show that carotenoids are vulnerable to degradation. It is especially their unsaturated structures that make them sensitive to heat, oxygen and light (Leskova et al., 2006). A study from Nigeria on heating of palm oil demonstrated that the amount of β-carotene declined with temperature increase and that the destruction of β-carotene was greater when the oil was heated continuously for 30 min at any given temperature (Mudambi & Rajagopa, 2006). Another study on heating red palm oil showed that all trans-β-carotene were almost lost after 20 min (Fillion & Henry, 1998). Thus given that local practices expose palm oil to sunlight and long duration cooking it is unlikely that palm oil consumption would have compensated for the lack of provitamin A in the local diet. However we would like to mention that intervention studies with red palm oil have found improved vitamin A status (Bhaskaram et al., 2003).

Besides our study there are only two other observational longitudinal studies we know of that examine the relationship between diet and the development of kwashiorkor: the Gopalan (1992) study of Indian children up to 10 years and Lin et al. (2007) study on Malawian children. While our study analyses diet with reference to a variety of food items, Gopalan’s study concentrated on differences in terms of protein-calorie consumption. Gopalan did not find evidence that there were any differences in the protein calorie intake between cases of marasmus and kwashiorkor. The study conducted by Lin et al. (2007) actually found that the daily intake of vitamin A equivalents was low among children who developed kwashiorkor, but not significantly lower than in the control group. The diets of the study population in these studies were monotonous and therefore to some extent resembled the diet in Bwamanda. The diet of Indian children was based on rice and millet, while the diet of the Malawian children was corn-based supplemented by small fish. The diet in Bwamanda consisted to a large extent of maize, cassava roots and cassava leaves.

Golden & Ramdath (1987) suggested that kwashiorkor results from oxidative stress and a general deficiency in protective mechanism that could reduce the oxidative damage with most of the protective pathways necessitating micronutrients. Before target specific antibodies have been produced, immune cells generate and release reactive oxygen species (Strobel, Tinz & Biesalski, 2007). These oxidative bursts have been shown to be important in the early phase of for example malaria infections. Oxidative burst can lead to haemolysis and cellular dysfunction (Isaksson et al., 2013). Since carotenoids can act as important antioxidants our findings support theories that relate kwashiorkor to oxidative stress and the importance of micronutrients in the protective pathways (Sergio, 1999; Strobel, Tinz & Biesalski, 2007).

A case-control study examined the antioxidant hypotheses by comparing the diet in siblings of children presenting with marasmus and children with kwashiorkor (Sullivan et al., 2006). It concluded that siblings of children with kwashiorkor consumed egg and tomatoes less frequently than children with marasmus did. We note that the differences in consumption between the two were minor.

Studies have questioned the hypothesis that antioxidant depletion causes kwashiorkor. In a controlled trial from Malawi, Ciliberto et al. (2005) assessed the efficiency of antioxidant supplementation in preventing kwashiorkor in children aged 1–4 years. The intervention arm received antioxidant powder containing riboflavin, vitamin E, selenium and N-acetylcysteine, while the control arm received a placebo of an identical looking powder. According to Ciliberto et al. (2005) the study showed that the supplementation of antioxidant powder did not prevent children from developing kwashiorkor. We noted that the study did not include supplementation of provitamin A and therefore does not address the efficiency of provitamin A in preventing kwashiorkor. The study did not provide baseline data about study participants’ diet. The study design does not, therefore, allow for examining how dietary variables might have influenced the treatment with antioxidant powder and the influence of such variables on the development of kwashiorkor.

As far as we know the Bwamanda study is, excepting Gopalan (1992) and Lin et al. (2007), the only observational longitudinal study that describes the diet of children who develop kwashiorkor and marasmus. The results from our study support the hypothesis that kwashiorkor is caused by oxidative stress, supporting the role that food containing carotenoids plays in the protective pathway. In Bwamanda the dietary sources of preformed vitamin A are limited and pro-vitamin A carotenoid constitutes a major source of vitamin A. The pro-vitamin A carotenoids including β-carotene, can through cleavage be converted into retinaldehyde (a form of vitamin A). Studies have shown that the bioavailability of provitamin A carotenoids is less than of preformed vitamin A (de Pee et al., 1995; de Pee et al., 1998). Our study proposes that the consumption of fruits plays a role in reducing the risk of developing kwashiorkor. In comparison with green leafy vegetables and carrots, fruits are also more effective in improving vitamin A status among children (de Pee et al., 1998).

Studies have shown that there is a relationship between diet and infection during the development of kwashiorkor and that recurrent infections contributes to hypo-albuminaemia and the development of oedema (Frood, 1971; Whitehead, 1977). We also know that vitamin A modulates many types of specific and non-specific immune system and those vitamin deficiencies have a negative impact on different types of immunity functions (Stephensen, 2001; Villamor & Fawzi, 2005). In this manner provitamin A can play a role in reducing the severity of a number of types of infections and we speculate that provitamin A played a role in reducing the risk of developing oedema among chronically malnourished children in Bwamanda.

The strength of our study is that in a large population-based study we have managed to examine the dietary diversity over a long period of children who developed kwashiorkor and marasmus. The study design has also some disadvantages. Given that information on diet was based on 24 h recall we were not able to determine food consumed in between the follow up period. Furthermore, we have limited information on the quantity of food consumed and we are therefore not in a position to determine the amount of pro-vitamin vitamin A consumed by the children. In our analysis we assumed that the food items reported at a given point in time were consumed up to the next survey round. Given that food consumption is characterised by individual variances this assumption represents a weakness. We also realise that our findings require confirmation, preferably in a large randomised trial that examines the development of kwashiorkor in a trial of carotenoids supplementation.

In conclusion, this is the first observational longitudinal study that demonstrates a relationship between diet diversity and the development of kwashiorkor. We have suggested that the consumption of a diet that includes food items containing carotenoids reduces the risk in children aged 10–71 months of developing kwashiorkor. Our findings should be considered as a support to ongoing efforts that aim at promoting a diverse agricultural and horticulture production and in this manner stimulate consumption of a more varied diet. In rural communities where there is a shortage of vitamin A rich food it is in particular important to promote increased production of vegetables and fruits rich in carotenoids including sweet potatoes and papaya.

Supplemental Information

Supplemental Information 1 Child with severe kwashiorkor

During a study in Bwamanda in 2013, this child was assessed according to nutritional status including occurance of oedema. The child on this photo was identified with kwashiorkor.

Click here for additional data file.

Supplemental Information 2 Pitting oedema

A Congolese doctor identifying pitting oedema on a child with kwashiorkor.

Click here for additional data file.

We would like to thank Roger Eeckels who participated in the initiation and conducting of the Bwamanda study.

Additional Information and Declarations

Competing Interests

Author Contributions

Human Ethics

Data Deposition

The authors declare there are no competing interests.

Hallgeir Kismul conceived and designed the experiments, performed the experiments, analyzed the data, wrote the paper, prepared figures and/or tables, reviewed drafts of the paper.

Jan Van den Broeck conceived and designed the experiments, performed the experiments, contributed reagents/materials/analysis tools, reviewed drafts of the paper.

Torleif Markussen Lunde conceived and designed the experiments, performed the experiments, analyzed the data, prepared figures and/or tables, reviewed drafts of the paper.

The following information was supplied relating to ethical approvals (i.e., approving body and any reference numbers):

Ethical approval for the Bwamanda study was granted by the University of Leuven’s Tropical Childcare Health Working Group.

As the principle investigator Jan Van den Broeck is the custodian of the Bwamanda dataset. Please contact Jan Van den Broeck for access to the dataset: Jan.Broeck@cih.uib.no.

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
