# Peer review of "Diet and kwashiorkor: a prospective study from rural DR Congo"

_PeerJ, doi:10.7717/peerj.350_

## Round 0.1 · original submission · Major Revisions

Some parts of the discussion and conclusions, outlined by the reviewers, should be better explained.

Reviewer 1 ·

Basic reporting

The manuscript might benefit from additional English language review.

Experimental design

No comments

Validity of the findings

No comments

Additional comments

Review

The authors analyzed the association between dietary intake and the advent of kwashiorkor and marasmus. To date, the causes of Kwashiorkor are not fully understood and many hypotheses have been formulated. Therefore, the subject of this paper remains relevant and important. The authors have used a rather unconventional analysis method that merits more explanation in the methods section. The manuscript might benefit from the presentation of more descriptive data from the cohort. I am at present unable to assess the quality of the paper due to the lack of clarity and additional results.

1)Abstract- last paragraph, two times ‘aged’ typo

2)Line 22: I would add here the non-result of the Ciliberto intervention study with anti-oxidants might be cited here as well (BMJ. 2005 May 14; 330(7500)) in replacement of the older reference.

3) line 42: 2 typo’s : ‘basic diet’ and ‘consists of mainly maize’

4) line 46: can this claim be supported by any reference.

5) line 49: delete ‘contacts,’

6) line 59: what is meant by ‘older’ children? Was this at 24 months (standing height instead of recumbent length)

7)line 61: the WHO-MGRS is unkown to me (MGRS?), are you referring to the 2006 WHO growth reference charts?

8) Line 64: replace ‘undertook’ by ‘conducted’

9) Line 65: I would refrain from using the term food frequency, because in the field of nutrition this is a common name for an instrument that recalls food intake over longer periods (weeks, months, years). ‘A single non-quantitative 24-hr recall’ might be the more appropriate term to use. (non-quantitative refers to the fact that quantities consumed were not measured).

10) Line 73: typo ‘did not’,

11)Please add a table that show the household and child characteristics stratified by Kwash/Marasmus/normal. Such descriptive table provides us with insight in other factors/covariates that differed between the groups. These may be the variables mentioned on line 76, but also additional ones.

12)Line 77-78: The cited analysis method needs a few lines more explanation on the concept and possible assumptions. As it is written now, it is rather unclear for the average reader. For instance, why a sample was taken from the study population, was this to validate the model? How was the sample taken? If 35 case-moments were selected, how many case-moments of kwashiorkor were there in the study base? What was done with cases of marasmus? Line 80: what is meant by ‘we took the dataset’?

13)Line 77-78: Please provide argumentation why this specific analysis method was used.

14)Line 83: shouldn’t it be a b/c ratio (c=35)?

15)Line 84: results in tables (1) should come under the ‘results’ section.

16)Line 85-section: Was the modeling exercise done on the sample (decribed one paragraph up)?

17)Line 88: Visual inspection to end up with a rather complex model on line 90 seems difficult. Can this step be explained better? How can one deduce this on sight? Wasn’t any software used to fit different models?

18)Line 94: I’m sure that an average reader not experienced in (statistical) modeling understands concepts like golden section and successive parabolic interpolation. Can a short explanation be added (maybe also with a reference to a standard work?).

19)Line 97: you announce two steps; for the sake of clarity, can you formally announce the second step?

20)Line 99: ‘at the last visit’ Do you mean the last visit before kwash/marasmus was found? If so, please add.

21)Line 101-103 represents the weakness of the study. Food and nutrient intake is characterized by an important intra-person variance. Inference on individual level based on this data would be impossible. Add to this that during infancy the diet can alter rapidly as a function of age and it becomes difficult to link the exposure data to the outcomes.

22)Line 110: typo in last sentence?

23)Line 114: typo first word


24)Line 121: was it significantly lower for cassava leaves in kwash children? How should we interpret the word ‘significantly’ in the text, no statistical test to compare the frequencies was done?


25)Line 133 I wonder why mango was not included in the PaSV indicator variable as it is richer in beta-carotene than papaya.

26)Line 136-137: rephrase, typo’s

27)Line 136: are all leafy vegetables added to this category? (Amaranth, Cassava leaves?)

28)Figure 1: I don’t understand how you can give 3 months incidence on age in 6-months intervals. Isn’t this just the prevalence by month of age?

29)Line 170: add ethical approval paragraph under methods.

30)Line 176: ‘chronically malnourished’ if stunting is intended here, please ‘stunting’ both terms are not the same.

31)Table 3: Can units be added in the table? It would be informative to get more summary crude data per case group in a few columns before the log odds estimate. Also providing P-values is recommended.

32)Figure 2: check first line of legend, something is not right.
Line 185:


33)Discussion: it might be useful to compare characteristics and diets between your study and those two others.

34)Discussion: I find the underscoring of palm oil as a source a bit troublesome. Intervention studies in different Sub-Sahara African countries showed significant increases in vitamin A concentrations in blood and breast milk. However, if your hypothesis holds, I don’t see any other reason than the ones cited. However, it might be more balanced to cite the intervention studies with red palm oil that improved vitamin A status as well.

35)Line 215: typo ‘oxiadative’

36)Line 217: typo ‘case-control’

37)Line 252: ‘population-based’

38)Line 256: ‘limited’ instead of ‘restricted’

39) Discussion: the discussion of the results on Marasmus are lacking

·

Basic reporting

No comment

Experimental design

This is an important and interesting study. As mentioned by the authors, this is the most important prospective study examining the relationship between dietary diversity and kwashiorkor.

In the background section, the authors should mention that there are major difficulties with the oxidative stress hypothesis. There is no oedema in other conditions associated with oxidative stress such as nuclear radiation, infections, exposure to oxidant drugs etc. In HIV infection there is a major oxidative stress, but there is an excess of non oedematous malnutrition among HIV patients who are severely malnourished. Also, there is no association between gene variants associated with anti oxidant protection and oedematous malnutrition. See:

Marshall KG, Swaby K, Hamilton K, Howell S, Landis RC, Hambleton IR, Reid M, Fletcher H, Forrester T, McKenzie CA. A preliminary examination of the effects of genetic variants of redox enzymes on susceptibility to oedematous malnutrition and on percentage cytotoxicity in response to oxidative stress in vitro. Ann Trop Paediatr. 2011;31(1):27-36.

The authors should also be careful re. the causality of the association they found. In the discussion, they should mention that as any observation study, this observed association may be due non causal. First, the association between intake of carotene rich foods and reduced risk of kwashiorkor may be due to an unidentified social or other confounding factor. Second, there are many other nutrients beyond beta carotene in fruit and vegetables which can have an effect on apparition of oedema. For instance, these foods have a high potassium content which may play a role in the origin of oedema.

So the authors should be more cautious in their conclusions and should mention the need of a prospective trial to confirm the hypothesis of a possible protective effect of beta carotene.

Validity of the findings

See above

Additional comments

The text should be edited and checked.

line 55 « feet », not « feets »
line 73 « did not » not « did no »
line 77 “meals” not “meas”.
Lines 127-132 Results difficult to follow. This information would be better presented with a figure.
Line 144. The high correlation reported between HFA and WHZ is quite uncommon. Both are usually only weakly correlated. Check this result which looks very unlikely. See: Richard SA, Black RE, Gilman RH, Guerrant RL, Kang G, Lanata CF, Mølbak K, Rasmussen ZA, Sack RB, Valentiner-Branth P, Checkley W; Childhood Infection and Malnutrition Network.Wasting is associated with stunting in early childhood. J Nutr. 2012 Jul;142(7):1291-6.
Line 168. meaning of mainfell ?
line 187 “extracted”
Lines 170 and following. Ethical approval is usually mentioned at the end of the methods section.
Line 206: “We suspect that if the study had examined the cumulative intake of vitamin A
equivalents over a longer period the difference in vitamin A equivalents, intake between children with kwashiorkor and marasmus might have been significant”. This is an unscientific statement. The opposite could happen. Delete.
Line 209 “deficiency”, not “defiance”.
Line 218 reference 23 is not about a case control study. Check.
beta carotene starts sometimes with a C. Make it upper or lower case all over the text.

Figure 1 : legend should give more detail. Not clear how the graph was made. Meaning of short vertical blue lines on the graph not given.

---

## Round 0.2 · Minor Revisions

The following minor comments should be included in the manuscript:
- Line 51. Please provide the two pertinent references.
- Lines 103-20. Use past tense intead of present tense.
- Please include in the manuscript the answer to the comment 24 from Reviewer 1.

·

Basic reporting

ok

Experimental design

ok

Validity of the findings

ok

Additional comments

No further comments, paper has improved and is much more clear.

---

## Round 0.3 · accepted · Accept

Congratulations for this interesting study